# Joint Performance of a Continuous Glass Fiber/Polypropylene Composite

**DOI:** 10.3390/polym15193942

**Published:** 2023-09-29

**Authors:** Shuai Jin, Liming Chen, Shaowei Zhu, Bing Du, Tao Liu, Xianbo Hou

**Affiliations:** 1College of Aerospace Engineering, Chongqing University, Chongqing 400030, Chinaclm07@cqu.edu.cn (L.C.);; 2State Key Laboratory of Coal Mine Disaster Dynamics and Control, Chongqing University, Chongqing 400030, China; 3School of Metallurgy and Materials Engineering, Chongqing University of Science and Technology, Chongqing 401331, China

**Keywords:** thermoplastic composite, joint performance, interface enhancement, mechanical properties

## Abstract

Thermoplastic composite structures possess superior properties compared with thermosetting composites, including recyclability and high damage tolerance. However, the poor adhesion properties of thermoplastic composites make their joining process challenging. In this research, three bonding techniques, namely adhesive, mechanical joining, and hybrid bonding, are investigated using lap shear specimens to evaluate their mechanical properties and failure modes. The stress distributions at the joints of the three bonding techniques are analyzed by numerical simulation. The findings demonstrate that hybrid bonding enhances the strength of composite joints, albeit at the expense of some stiffness due to the presence of an open hole. This method is particularly suitable for applications that necessitate robust connections requiring high strength.

## 1. Introduction

Recently, there has been a growing interest in continuous fiber-reinforced thermoplastic composites due to their remarkable properties [1,2,3]. These composites exhibit high damage resistance [4,5], recyclability [6,7,8], ductility [9,10], repairability [11], and recoverability [12] when compared with conventional thermosetting composites. As a result, they hold great potential as important materials in the aerospace [13,14], automotive [15], and marine industries [16].

Joining is a critical aspect in the manufacturing of continuous fiber-reinforced composites, as it plays a vital role in the production of large composite structures [17]. However, it is often considered a weak point in composite structures. Extensive research has been conducted on the joining techniques for composites, with a particular focus on joining problems in honeycombs [18,19], lattice structures [20,21,22,23], foams [24], and origami [25,26,27]. For thermoset composite structures, bonding [27] and metal bolt connections [28] are commonly employed. In contrast, bonding thermoplastic composites is more challenging compared with thermoset composites due to their poor adhesive properties. However, one advantage of thermoplastic composites is that they can be cured twice, which opens up new opportunities for developing joining techniques. Researchers are exploring innovative methods to effectively bond thermoplastic composites by taking advantage of their unique properties.

To prepare structures with corrugated cores, Schneider et al. [10] used the face sheet-core co-curing preparation process for connection. With this joining method, the sandwich structure was successfully fabricated; however, the material was discontinuous between unit cells, resulting in resin-rich zones at the transitions that compromised the overall mechanical performance. Hu et al. [29] used hot-pressing for connecting the sandwich panel and truss core. In the flatwise compression tests, the structure exhibited rod buckling/fracture or local panel damage without core–skin debonding. This indicates the feasibility of the joining method. However, specific tooling is required to fix the structure and prevent deformation during hot-pressing, since the entire structure is heated for bonding. Du et al. [30] studied the resistance welding of glass fiber-reinforced thermoplastic composite. This method achieved good joint strength while avoiding the overall heating of the structure during processing. However, the welding process inevitably introduced metallic impurities (serving as conductors during welding). Further exploration, optimization, and comparative study of composite connections remains high value due to the increased demand for composites in industries such as the automotive industry. In addition, a great advantage of thermoplastic composites is recyclability [6,7,8]. However, common mechanical connection techniques (using metal bolts) introduce additional compounds that affect the recyclability of the composites. Overcoming this challenge remains an issue.

The present study investigates three different bonding techniques: (1) Adhesive bonding. This traditional method is used as a control group. (2) Mechanical connection. This is different from the common mechanical connection using metal bolts, this paper used the same material as the composite matrix as a stick for connection. (3) Hybrid connection. This is a combination of the first two connection methods. Through a comparative analysis of their mechanical properties and failure modes via experiments and simulations, the advantages and disadvantages of these three connection methods are examined.

## 2. Materials

This section introduces the materials used in the experiments and the implementation procedures of the three joining methods.

### 2.1. Preparation of Materials

The geometry of single lap shear specimens refers to GB/T 33334-2016 [31]. The material properties of glass-fiber/polypropylene (GF/PP) prepreg provided by KINGFA Technology Co. are listed in Table 1. The GF/PP laminate is laid for [0°/90°]_2s_. The GF/PP materials were fabricated via the hot-press method with the process parameters shown in Figure 1a. The hot-pressing procedure and the machine are shown in Figure 1b. After demolding, a strip with a length of 100 mm × 25 mm was cut with a CNC engraving machine and the edge of the specimen was polished with fine sandpaper to remove burrs. The polished strip of laminate was put into an ultrasonic cleaner for 5 min and then in an oven at 60 °C for 10 min.

### 2.2. Adhesive Bonding

Adhesive bonding is the most common joining technique, especially for thermoset composites. For the thermoplastic composites studied in this work, this process serves as a reference method for comparison. The specific process for the adhesive-bonding specimen is as follows. Firstly, 320# sandpaper is used to polish the bonding area to remove the release agent and increase roughness. Secondly, special wiping paper with alcohol is used to further clean the area to be bonded and remove dust and stains from the surface. Afterward, the strips are placed in an oven at 60 °C for 5 min. Thirdly, the surface to be bonded is evenly brushed with a surface polarizer and dried at room temperature for 15 min. Fourthly, the strips are coated with MS1937 modified silane glue (Tonsan Adhesive, Inc., Coral Gables, FL, USA) and cured at room temperature for more than 3 days with pressure. The uniaxial tensile properties of MS1937 were tested with the standard GB/T528-2009 [32], and the stress–strain curve is shown in Figure 2. The specific parameters of MS1937 are shown in Table 2, where RH indicates the relative humidity. The elongation at failure and tensile strength were tested according to the standard GB/T528-2009 [32]; the shear strength was tested according to the standard GB/T7124-2008 [33].

### 2.3. Mechanical Bonding and Hybrid Bonding

Traditional mechanical fastening generally uses metallic bolts or bolts made from other materials, which inevitably introduces foreign compounds that compromise the recyclability of the composites. The mechanical joining in this work utilizes PP sticks, which are the same material as the composite substrate. Thus, no foreign compounds are introduced, alleviating concerns over the effects of foreign compounds on material recyclability.

Hybrid bonding combines this mechanical joining method with adhesive bonding, aiming to achieve higher joint strength.

This section introduces the specific procedures and details of the two methods. The three types of fabricated bonding specimens are shown in Figure 3. Figure 3a shows the schematic diagram and a photo of the adhesive method. Figure 3b shows the schematic diagram and a photo of the mechanical bonding. A PP glue-stick-based hot-riveting process was used to make the mechanical bonding single-lap specimen. The process was completed as follows: Firstly, the relative position of the strips is fixed to form a single-lap preform. Then, the single-lap joint preform is drilled with a hole with a diameter of 5 mm using the CNC engraving machine. Thirdly, the PP glue stick (with the properties shown in Table 3) is cut into small sections with lengths of about 15 mm that are inserted into the prefabricated hole with 5–8 mm at both ends. Finally, the ends of the PP glue stick are melted using an electric soldering iron and then pressed with a metal plate to form a “hat-shaped” area to prevent the glue stick from falling off. The material parameters of the PP are from reference [34]. Hybrid bonding is a combination of adhesive and mechanical bonding. Figure 3c shows the schematic diagram and a photo of the hybrid bonding. The hybrid bonding specimen was cured and molded under the same temperature and humidity environment as the adhesive one and prepared under the same process as the mechanical bonding.

## 3. Experiment and Simulation

This section introduces the testing and simulation methods for the joined structures.

### 3.1. Experimental Setup

The single-lap specimens were tested on a SANS CMT5105 (100 kN) electronic universal testing machine according to ASTM D5868-01 [35]. As shown in Figure 4, the width *W* of the specimen is 25 mm, the length of a single laminate *L* is 100 mm, and the overlap area *A* is 625 mm^2^. The two ends of the specimen are protected with GF/PP laminate to prevent damage at the grips of the machine and to ensure the initial loading was vertical. The loading speed was 1 mm/min. At least three specimens were tested for each bonding method. The single-lap shear strength *σ_shear_* is
(1)$$\sigma_{shear}=\frac{P_{\max}}{A}$$
where *P*_max_ is the maximum load.

### 3.2. Numerical Simulation

ABAQUS/Explicit was used to simulate the single-lap shear test of the three bonding techniques. The composite laminate and PP glue stick were meshed as C3D8R (three-dimensional eight-node linear with reduced integration elements) finite elements and the adhesive layer was meshed as COH3D8 (eight-node three-dimensional cohesive elements) finite elements. The mesh size of all elements was 1 mm. A tie constraint was applied between the laminate and adhesive layer. As shown in Figure 5, each end of the laminate was coupled to a reference point. One end was all fixed and another was fixed except in the x-direction, where positive displacement was applied. The surface-to-surface contact was set between the middle part of the PP glue stick and the inner surface of the hole of the laminate, between the end part of the PP glue stick and the top and bottom surfaces of the laminate, and between the inner surfaces of the two laminates. The contact attribute was set to hard contact, and the friction coefficient was set to 0.2.

The material properties of GF/PP are listed in Table 1. Three-dimensional Hashin damage initiation and stiffness degradation criteria were considered. PP is regarded as an isotropic elastoplastic material; the ductile damage criterion was adopted for failure and the material parameters were obtained from reference [22]. The cohesive parameters of the adhesive layer material are determined according to the data provided by the manufacturer and the performance parameters of similar modified silane adhesives, as shown in Table 4. In this table, *E*_nn_ represents the Young’s modulus of the MS1937 adhesive in the normal direction. *E*_ss_ and *E*_tt_ represent the shear modulus of the MS1937 adhesive in two local shear directions. $t_{n}^{0}$, $t_{s}^{0}$, and $t_{t}^{0}$ represent the peak values of the stress when the deformation is normal to the interface, in the first shear direction, and in the second shear direction, respectively. The quadratic nominal stress criterion (Equation (2)) was adopted as the damage initiation of the adhesive layer. 

The B-K energy criterion based on mixed modes is adopted for the damage evolution of the adhesive layer. $G_{I}/G_{II}/G_{III}$ are the strain energy release rates and $G_{Ic}/G_{IIc}/G_{IIIc}$ are critical strain energy release rates corresponding to type I, type II, and type III cracks, respectively. *η* is the damage factor, with a value of 0.5–2.0, *D* is the stiffness reduction factor, which can be calculated by Equation (3), and the glue layer unit is completely invalid and deleted when *D* = 1.
(2)$${\left\{\frac{\langle t_{n}\rangle }{t_{n}^{0}}\right\}}^{2}+{\left\{\frac{t_{s}}{t_{s}^{0}}\right\}}^{2}+{\left\{\frac{t_{t}}{t_{t}^{0}}\right\}}^{2}=1$$
(3)$$D=\frac{G_{I}+G_{II}+G_{III}}{G_{Ic}+\left(G_{IIc}-G_{Ic}\right){\left[\left(G_{II}+G_{III}\right)\left(G_{I}+G_{II}+G_{III}\right)\right]}^{\eta }}$$

## 4. Results and Discussion

This section presents the experimental results and compares the three connection techniques through simulation.

### 4.1. Adhesive and Mechanical Bonding

The load–displacement curve of the adhesive bonding specimen is presented in Figure 6a. Figure 6b shows the fracture surface of the specimen after the experiments. During the linear elastic stage, the bonding strength between the adhesive and the laminate surpasses the internal stress of the adhesive layer. Upon reaching the peak load, the internal stress of the adhesive layer meets the criterion for damage initiation. Localized failure then initiates, causing the failure area of the adhesive layer to progressively expand until ultimate failure occurs.

The adhesive bonding specimens exhibited an average peak load of 1070 N, an average initial stiffness of 1757 N/mm, and an average of the displacement at the peak load of 0.82 mm. Examination of the fracture surface of the joint revealed that the primary failure mode was adhesive failure. This suggests that the GF/PP laminate demonstrates improved interface performance with the adhesive following surface polarization.

By numerical simulation, peeling stress and shear stress in the adhesive layer at peak load are illustrated in Figure 7. Here, the normalized distance is defined to represent the position, as shown in Figure 7a. Figure 7b,c shows the peel stress and shear stress at peak load. Notably, the highest stress occurred at the end of the adhesive layer, coinciding with the loading where failure initiates. Relatively speaking, the middle of the adhesive layer was under less stress. Such a stress distribution causes delamination to always first occur at the ends of the adhesive layer. However, improvement is difficult. Some possible approaches such as adjusting the adhesive layer thickness may be useful but are challenging to implement in manufacturing.

The load–displacement behavior of the mechanical bonding specimens is represented in Figure 8. The average peak load for these specimens was 489 N, with an average initial stiffness of 642 N/mm and an average displacement at the peak load of 3.09 mm. Once the peak load was reached, the laminate joint underwent deflection and the PP glue stick experienced plastic deformation until failure occurred.

In contrast to traditional bolt bonding methods (using metal bolts), the proposed PP-glue-stick-reinforced bonding process in this study aims at achieving weight reduction and recyclability. Since the PP sticks and the composite matrix are made of the same material, they do not need to be separated during recycling, thus avoiding potential impacts on recyclability (if metal bolts were used for the connection, the metal bolts and composite would need to be separated during recycling). However, due to the absence of pre-tension applied to the bonding surface, the two laminates tended to separate during the deformation process. This is also a drawback of using PP sticks for mechanical connection. To use the same material as the matrix of composites, the performance of the sticks is sacrificed. The ideal solution would be to also make the PP sticks fiber-reinforced, but this is technically difficult to achieve and difficult to provide end constraints.

Figure 9 illustrates the finite element analysis results at the peak load, revealing the stress concentration at the edge of the laminate hole. Figure 9a shows the in-plane normal stress S11 in the outer layer. Figure 9b shows the in-plane normal stress S22 in the outer layer. Figure 9c shows the in-plane shear stress S12 in the outer layer. Figure 9d shows the in-plane normal stress S11 in the cross section. Figure 9e shows the in-plane normal stress S22 in the cross section. Figure 9f shows the in-plane shear stress S12 in the cross section. Figure 9g shows the Mises stress in the cross section in the PP stick. Importantly, the stress levels in this region remained below the material’s tensile strength. Meanwhile, the middle portion of the PP glue stick underwent plastic deformation under both tensile and shear forces. This also indicates that the main reason for the joint detachment is the shear failure of the PP pins. Although the composite plate was subjected to the combined action of shear and normal stresses, the stresses were not sufficient to cause its failure. For this failure mode, enlarging the radius of the holes and of the PP sticks may be a good solution; however, since drilling holes interrupts fibers and degrades the performance of the composite itself, how to balance the two still needs further study.

### 4.2. Hybrid Bonding

Hybrid bonding refers to the combination of adhesive bonding and mechanical bonding processes. Figure 10 shows the load–displacement curve of hybrid bonding lap shear specimens. The load–displacement curve of the hybrid bonding specimen can be divided into three distinct stages. Firstly, during the linear elastic stage, both the adhesive layer and PP glue stick undergo elastic deformation. Once the load reaches its peak, the adhesive layer starts experiencing local damage, whereas the PP glue stick undergoes plastic deformation, resulting in a gradual decrease in load. Finally, the adhesive layer is completely destroyed, leaving the PP glue stick to solely bear the load.

The average peak load for the hybrid bonding specimen was 1352 N, with an average initial stiffness of 1532 N/mm and an average displacement at the peak load of 1.1 mm. 

Figure 11 shows the stress distribution in hybrid specimens at the peak load point. Figure 11a shows the in-plane normal stress S11 in the outer layer. Figure 11b shows the in-plane normal stress S22 in the outer layer. Figure 11c shows the in-plane shear stress S12 in the outer layer. Figure 11d shows the in-plane normal stress S11 in the cross section. Figure 11e shows the in-plane normal stress S22 in the cross section. Figure 11f shows the in-plane shear stress S12 in the cross section. Figure 11g shows the Mises stress in the cross section in the PP stick. This figure demonstrates that, in hybrid bonding, the overall internal stress of the laminate is increased compared with that in mechanical bonding. This is also the reason why this type of connection has greater strength. The adhesive layer transferred part of the load through shear stress, sharing the load on the PP sticks so that this hybrid connection could withstand greater loads.

The comparison of the mechanical properties of the three bonding specimens is presented in Figure 12. Hybrid bonding exhibits the highest bonding strength, followed by adhesive bonding; mechanical bonding exhibits the lowest strength. The specific values are 2.1 MPa, 1.6 MPa, and 0.7 MPa, respectively. It is worth noting that the strength of the hybrid bonding joint is slightly lower than the combined strength of adhesive bonding and mechanical bonding. This can be attributed to the reduced performance of the adhesive layer after the hole is opened. In terms of stiffness, adhesive bonding demonstrates the highest value at 1757 N/mm, followed by hybrid bonding at 1449 N/mm, and mechanical bonding has a stiffness of 642 N/mm. This discrepancy is primarily due to the presence of openings in the composite material. Additionally, the inclusion of the PP glue stick, which has a significantly lower stiffness compared with the composites, impacts the overall stiffness of the joint.

Overall, for PP composites, mechanical joining (using PP sticks) does not introduce additional compounds, which greatly reduces the potential negative impact on the recyclability of the joining materials; however, the strength is relatively low. Adhesive bonding offers moderate strength but introduces other compounds. Hybrid joining achieves higher strength but, like adhesive bonding, introduces foreign compounds. Additionally, the processing for hybrid joining is the most complex among the three methods. In future application in aerospace, the automotive industry, and other areas, these three techniques will have their own pros and cons, and the most suitable technique should be selected based on the specific requirements.

This study also reveals that both the mechanical and hybrid connections exhibit an issue stemming from hole drilling, which interrupts fibers and degrades the performance of continuous fiber composite materials. However, smaller holes would also limit the size of the PP sticks, restricting their ability to withstand high loads. Further research should focus on optimizing the hole dimensions or altering fiber orientations, which would be achievable through advanced manufacturing techniques such as 3D printing, to mitigate the detrimental impacts of perforations on material properties.

## 5. Conclusions

Bonding techniques including adhesive, mechanical, and hybrid bonding were experimentally and numerically investigated using lap shear specimens in terms of mechanical properties and failure modes. Wherein, the mechanical connection employed a self-tapping method, that is, using sticks made of the same material as the composite matrix to connect. The hybrid connection was a combination of the mechanical connection and bonding. Compared with traditional mechanical connection (using metal sticks), the advantages of these methods are that they do not introduce additional substances and thus have less impact on the recyclability of the composites. Through experimental and simulation studies, the conclusions are as follows:In the experiments, hybrid bonding has the highest bonding strength, followed by adhesive bonding, and the worst is mechanical bonding.The strength of the hybrid bonding joint is slightly less than the sum of the adhesive bonding and the mechanical bonding. The main reason for this is that the performance of the adhesive layer is reduced to a certain extent after the hole is opened.The stiffness of hybrid bonding is less than adhesive bonding. This is mainly due to the effect of openings in the composite. The filled PP glue stick affects the joint stiffness due to its much lower stiffness than the composite material.

Overall, the findings of this study contribute to the understanding of bonding techniques in composite applications. In particular, it contributes to connecting materials without affecting the recyclability of the composite material. Further advancements and refinements of these techniques can unleash the potential of composites, resulting in improved performance and expanded applications across various industries, including aerospace and the automotive industry. For future research, more quantitative studies are needed, such as optimization of hole dimensions, etc. In addition, conducting complete recycling experiments is also expected.

## Figures and Tables

**Figure 1 polymers-15-03942-f001:**
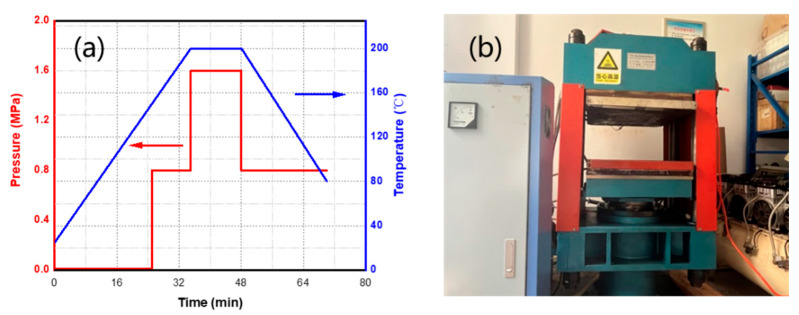
Preparation of materials: (**a**) processing parameters of hot-pressing and (**b**) the hot-pressing machine.

**Figure 2 polymers-15-03942-f002:**
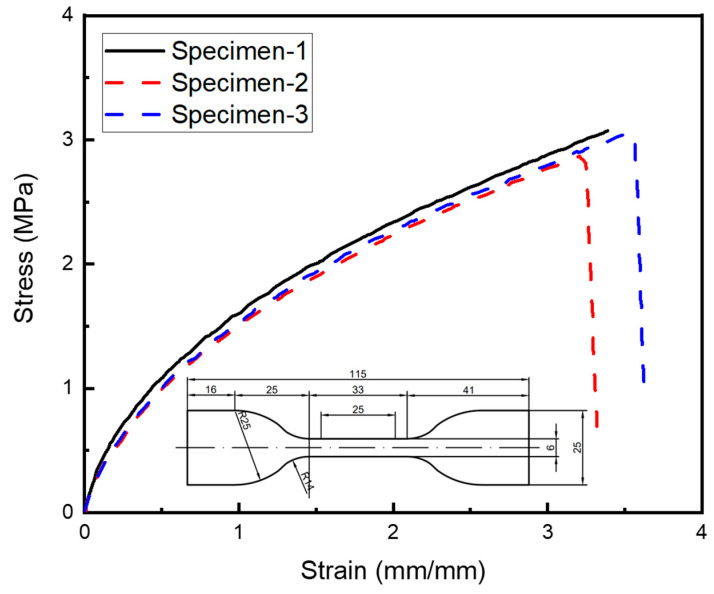
Stress–strain curves of MS1937 adhesive under uniaxial tension.

**Figure 3 polymers-15-03942-f003:**
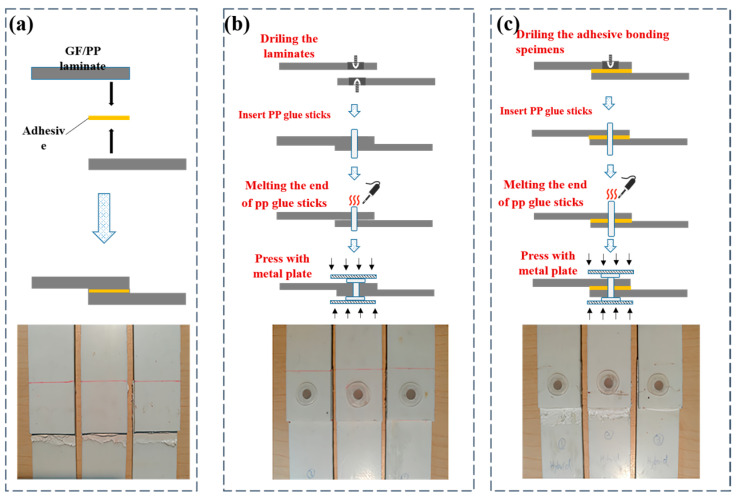
Comparison of three kinds of single lap shear specimens: (**a**) Adhesive, (**b**) Mechanical bonding, (**c**) Hybrid bonding.

**Figure 4 polymers-15-03942-f004:**
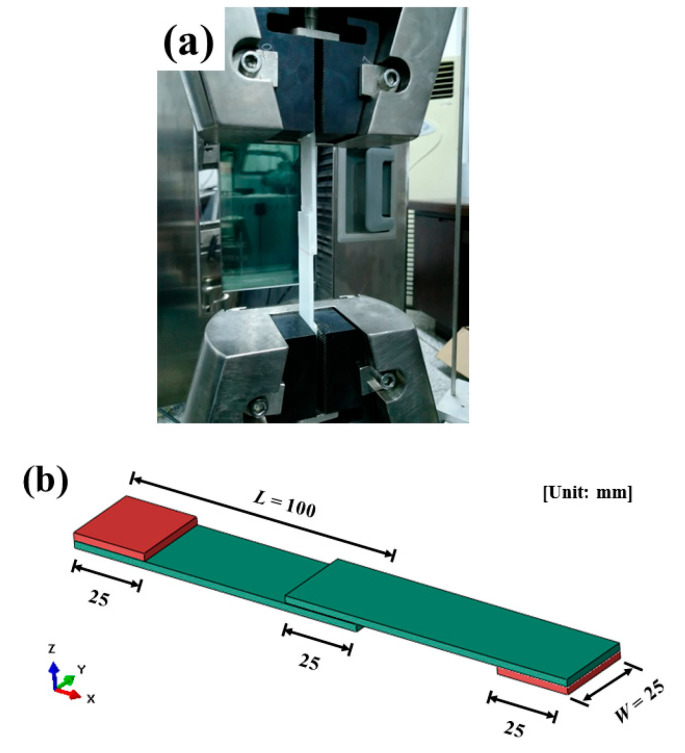
Single-lap shear test of mechanical bonding: (**a**) experimental setup and (**b**) specimen.

**Figure 5 polymers-15-03942-f005:**
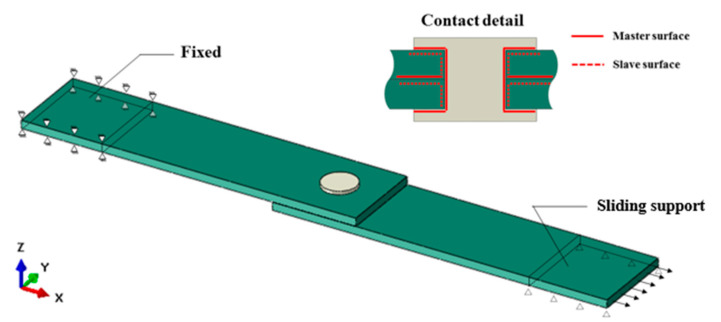
Finite element modeling, taking mechanically bonded specimens for an example.

**Figure 6 polymers-15-03942-f006:**
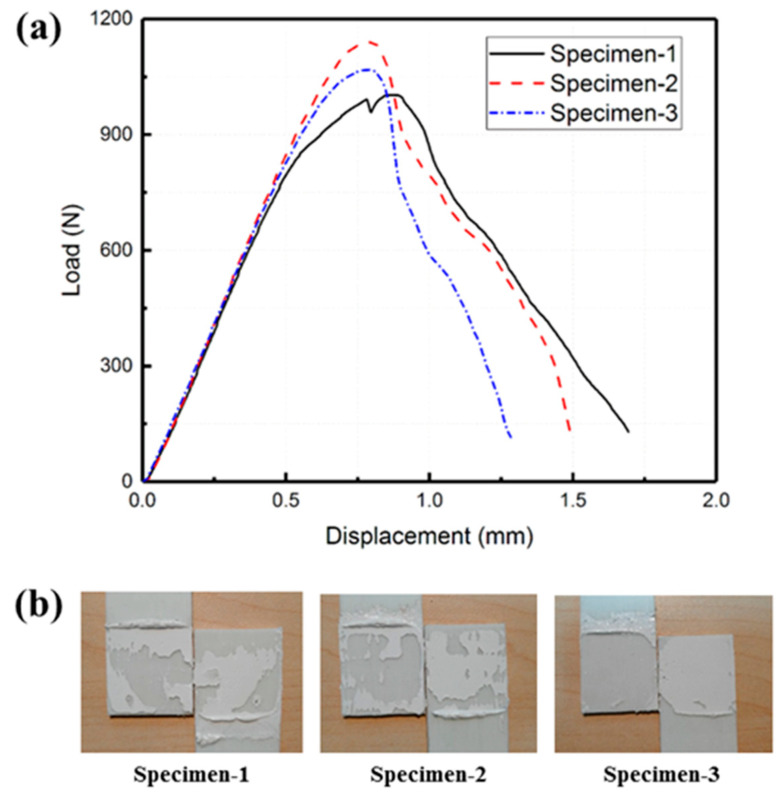
Adhesive bonding lap shear specimen: (**a**) load–displacement curve and (**b**) fracture surface.

**Figure 7 polymers-15-03942-f007:**
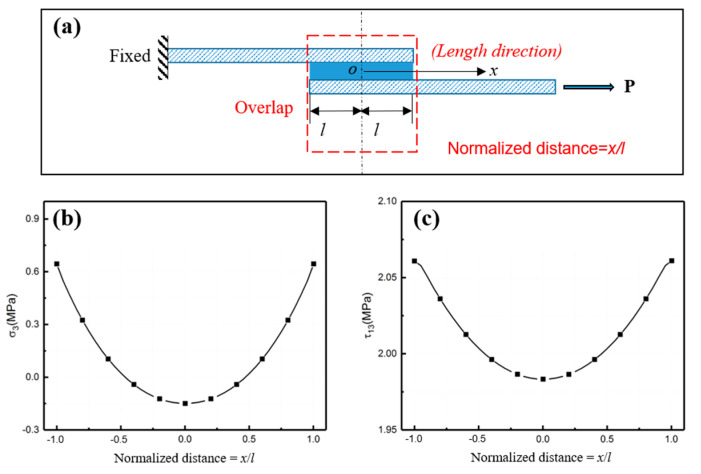
The stress distribution in adhesive layer. (**a**) Conceptual diagram of normalized distance. (**b**) Peel stress and (**c**) shear stress at peak load.

**Figure 8 polymers-15-03942-f008:**
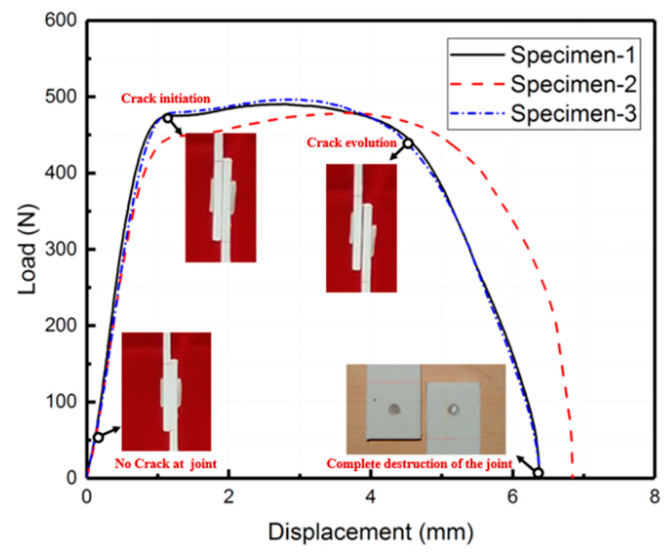
The load–displacement curve of mechanical bonding lap shear specimens.

**Figure 9 polymers-15-03942-f009:**
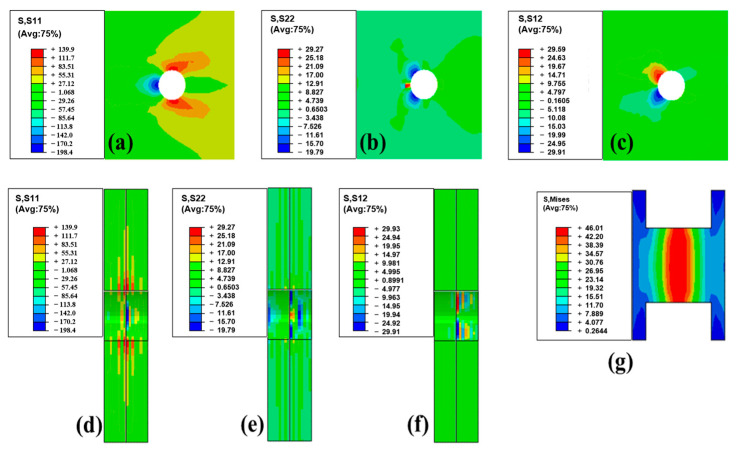
Stress distribution in mechanical bonding specimens (at peak load point): (**a**) In-plane normal stress S11 in the outer layer. (**b**) In-plane normal stress S22 in the outer layer. (**c**) In-plane shear stress S12 in the outer layer. (**d**) In-plane normal stress S11 in the cross section. (**e**) In-plane normal stress S22 in the cross section. (**f**) In-plane shear stress S12 in the cross section. (**g**) Mises stress in the cross section in the PP stick.

**Figure 10 polymers-15-03942-f010:**
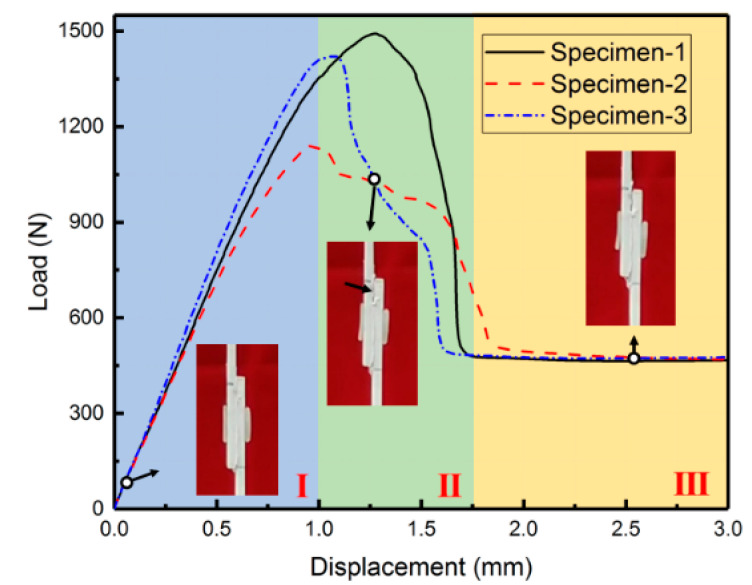
The load–displacement curve of hybrid bonding lap shear specimens.

**Figure 11 polymers-15-03942-f011:**
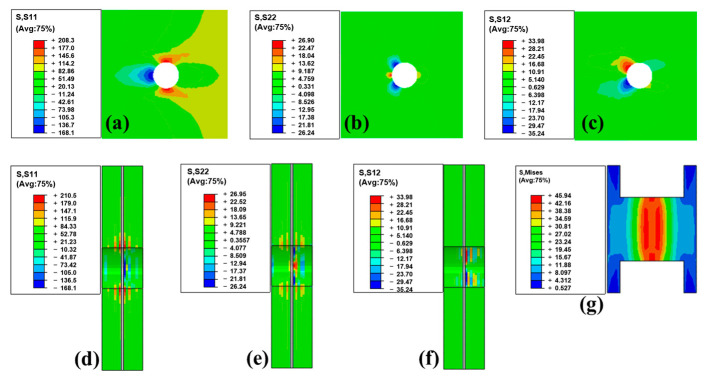
Stress distribution in hybrid specimens (at peak load point): (**a**) In-plane normal stress S11 in the outer layer. (**b**) In-plane normal stress S22 in the outer layer. (**c**) In-plane shear stress S12 in the outer layer. (**d**) In-plane normal stress S11 in the cross section. (**e**) In-plane normal stress S22 in the cross section. (**f**) In-plane shear stress S12 in the cross section. (**g**) Mises stress in the cross section in the PP stick.

**Figure 12 polymers-15-03942-f012:**
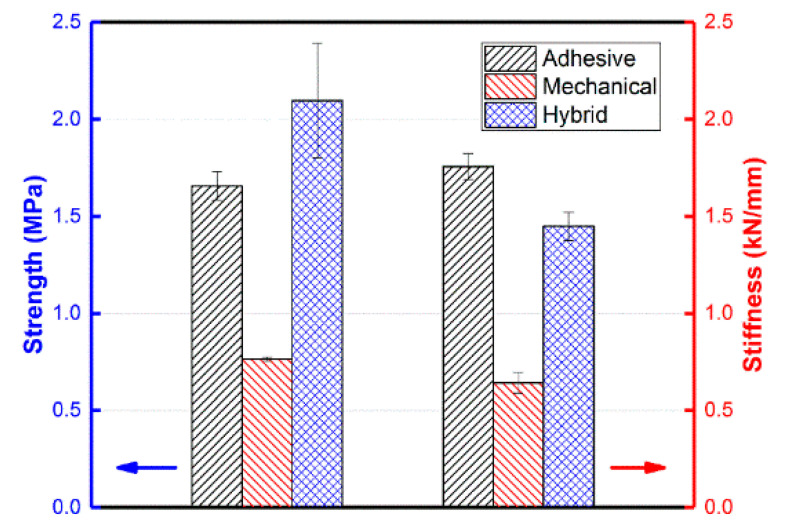
Comparison of the mechanical properties of the three bonding specimens.

**Table 1 polymers-15-03942-t001:** Properties of the GF/PP prepreg provided by KINGFA Composites.

Property	Value
Density	1.5 g/cm^3^
Fraction of fibers (weight)	60%
Heat deflection temperature, 0.45 MPa	155 °C
Thickness	0.3 mm
Young’s modulus in the longitudinal direction	28 GPa
Young’s modulus in transverse direction	3.2 GPa
In-plane shear modulus	946 MPa
In-plane Poisson’s ratio	0.064
Longitudinal tensile strength	750 MPa
Longitudinal compressive strength	160 MPa
Transverse tensile strength	15 MPa
Transverse compressive strength	50 MPa

**Table 2 polymers-15-03942-t002:** Adhesive properties of MS1937 provided by Tonsan Adhesive, Inc.

Property	Value
Density	1.45 g/cm^3^
Tack free time (@25 °C, RH50%)	5–20 min
Cure speed (@25 °C, RH50%)	4 mm/24 h
Elongation at failure (GB/T528)	>200%
Tensile strength (GB/T528)	3.0 MPa
Shear strength (GB/T7124)	2.3 MPa

**Table 3 polymers-15-03942-t003:** Properties of the PP.

Property	Value
Density	0.90 g/cm^3^
Melt flow rate (230 °C, 2.16 Kg)	19 g/10 min
Heat deflection temperature,0.45 MPa	120 °C
Bending strength	41.5 Mpa
Bending modulus	1.72 Gpa

**Table 4 polymers-15-03942-t004:** Cohesive parameters of the MS1937 adhesive.

*E*_nn_ (MPa)	*E*_ss_ (MPa)	*E*_tt_ (MPa)	$t_{n}^{0}$ (MPa)	$t_{s}^{0}$ (MPa)
1.9	1	1	3	2.3
$t_{t}^{0}$ (MPa)	*G_Ic_* (N/mm)	*G_IIc_* (N/mm)	*G_IIIc_* (N/mm)	$t_{t}^{0}$ (MPa)
2.3	1	1	1	2.3

## Data Availability

The data presented in this study are available on request from the corresponding author.

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
