# Peer review of "Joint Performance of a Continuous Glass Fiber/Polypropylene Composite"

_polymers, 2023, doi:10.3390/polym15193942_

Round 1

Reviewer 1 Report

This manuscript presents an experimental and numerical study about the mechanical performance of three types of connection systems between thermoplastic GFRP composites, namely adhesive bonding, mechanical bonding and hybrid bonding. The study is framed within a very relevant topic in the literature, as it addresses one of the main concerns regarding the use of thermoplastic composites for structural and non-structural applications.

Overall, the paper is well structured and written in proper English (minor typos were found, see below). The description of the experimental and numerical studies, as well as the discussion of the results obtained are clear and easy to follow. The quality of the figures needs to be improved. Please find in the “Comments” section below the list of detailed remarks made to the manuscript.

Based on the reasons stated above, the reviewer recommends the publication of this manuscript in Polymers journal, after minor revisions.

 Comments:

1.        The abstract should clarify that the research comprises both experimental and numerical studies.

2.        Did the joint techniques tested in this study been employed in other studies? If so, how do these results compare with those available in the literature? How do they compare to other techniques already in use in terms of ease of application, cost, capacity for recycling after end-of-life?

3.        Line 53: the notation “GF/PP” is first used here, hence it should be defined here as well (glass-fibre/polypropylene (GF/PP))

4.        Line 55: “[…] with the process parameters shown in Figure 1(a).”

5.        The multiplication sign should be corrected to “×” instead of using an asterisk (*) (see line 56 and 59 for example).

6.        Line 59: Before mentioning the overlap area, the type of specimen/geometry should be defined (i.e. mention that is a single lap shear specimen); this information only appears in the following section 3.1.

7.        The data provided in Table 1 and Table 2 derive from product catalogues, right? The reference of the datasheet should be provided in the references.

8.        Do you have any information about the thermophysical properties of the composite and of the adhesive/PP glue stick?

9.        Table 1: in the units of density, the number 3 should be in superscript (g/cm3).

10.     The resolution of the figures need improving. Moreover, the letters of some plots are very small and therefore difficult to read (see Figure 1, Figure 2 (tensile specimen dimensions), Figure 9, Figure 11). The indication of the master and slave surfaces is not clear in Figure 5 due to the lack of resolution of the figure.

11.     Legend of Figure 1: “Preparation of materials: (a) processing parameters of hot-pressing, and (b) hot pressing machine”.

12.     Line 66: “[…] specimen is as follows. Firstly, […]”

13.     Line 72: the authors should specify what is the brand/manufacturer of the adhesive used.

14.     Figure 2: was the adhesive tested by the authors or were these results provided by the manufacturer? This should be clarified in the text.

15.     Table 2: “elongation at break” doesn’t sound right. Do you mean elongation at failure?

16.     Table 2: the notations (RH (relative humidity), GB/T528 and GB/T7124) should be defined in the text.

17.     Line 86-87: The authors mention that the properties of the PP glue stick were provided by the manufacturer but don’t quantify them in the text (yet the properties of the adhesive used for adhesive bonding are listed).

18.     Line 100: “The two ends of the specimen are protected with GF/PP laminate […]” – right?

19.     Line 101: “[…] to prevent damage at the grips of the machine […]”

20.     Line 101: what to you mean by “ensure the initial vertical loading”?

21.     Line 102: “At least 3 specimens are tested […]”, instead of “carried out”.

22.     Section 3.2: it should be clarified which type of connections were numerically simulated? Was the adhesive connections also modelled?

23.     Line 109: “[…] C3D8R finite elements […]”. Also, what were the dimensions of the elements/mesh? What type of analysis was conducted?

24.     Instead of “moved supper”, “sliding support”?

25.     Line 125: “[…] nominal stress criterion (Equation (2)) […]

26.     The parameters presented in table 3 need to be defined (either on the text, on the table, or on the legend of the table).

27.     Line 142-143: “average stiffness” – it is actually the “average initial stiffness”. Also very this in the following sections (Lines 156 and 181-182).

28.     Line 143: “average failure maximum displacement” – do you mean the average displacement at failure? Or the maximum displacement attained? What does the value 1.1mm correspond to? Because it does not seem to be the average of the displacement at the peak load. Did failure occur at the maximum load? Also verify this issue in the following two sections (Line 157 and 182)

29.     Line 149: “[…] location of the initial failure” – do you mean the loading where failure initiates?

30.     Figure 7: where these results obtained from the numerical models? This should be mentioned.

31.     Lines 160-162: the authors mention that one of the benefits of the PP glue stick solution is its recyclability. In practice, would this recyclability be feasible? The quantity of PP glue is very low compared to that of the composite material; can these two components (pre-preg and PP glue stick) be easily separated and recycled (separately)? Is this solution practical and economically feasible?

32.     Can these 3 connection techniques be easily implemented in larger scale composites? The disadvantages of each technique should also be compared.

33.     Figure 8: it is not clear what the circles in the curves/figures represent. It would be useful if some legend was added near the figures/relevant points of the curve.

34.     Figure 8: the legend should read “load-displacement curve of mechanical bonding”, instead of “adhesive bonding”. Also “specimens” instead of specimen”.

35.     Figure 9: same comment as in figure 8.

36.     Line 195-196: in what way does the presence of a hole affect the performance of the adhesive? This is not clear. The hole is small compared to the bonded area, and the is made prior to the application of the adhesive. How could the loss of stiffness provided by the hybrid connection be counteracted? How could it be improved to allow the adoption of this technique rather than the adhesive or mechanical connections?

37.     Line 206: “[…] numerically investigated in lap shear specimens […]” (instead of “by”).

Author Response

Thank you for your comments. Please find our response in the attached document.

Reviewer 2 Report

Review: polymers-2622027

Title: Joint performance of continuous glass fiber/polypropylene composite

The paper deals with the comparative analysis of mechanical properties and failure modes of three different bonding techniques.

The topic of the article is interesting, but the author did not sufficiently highlight the complexity of the issues. The information is presented in a logical way and the explanations and conclusions are good, but the novelty of this work should be highlighted.

Introduction:

How the present study differs from the previously published study? In the introduction, some bibliographic references are listed without going into significant details for the work. I suggest a presentation of references oriented on results obtained in correlation with the current study.

Materials &method:

Fig 1a - requires quality improvement.

2.2. Adhesive bonding

Was the uniaxial tensile load-displacement curve of MS1937 determined experimentally by the authors? If so, what standard was used to determine the elastic properties of the adhesive?

2.3. Mechanical bonding and hybrid bonding

Instead of describing the stages of obtaining the three types of joints, the authors could introduce a scheme with the principle of obtaining joints. It would be much easier and intuitive for readers. In addition, from Fig.3 it is difficult to differentiate joints b and c.

3. Experiment and simulation

3.1. Experimental setup

The minimum of five lap shear samples should not have been tested?

3.2. Numerical simulation

-What does I means in C3D8R elements? -What does means in COH3D8. The author should use the general description of elements (number of nodes, number of integration points, reduced integration or not)

4. Results and Discussion

Figure 9 is not legible - the legend is not understood.

The number of tested samples is not relevant enough, for which ANOVA type statistical analysis cannot be applied. Thus, it is difficult to establish whether the obtained phenomena characterize all such connections or constitute a case.

Please highlight how the work advances or increments the field from the present state of knowledge and provide a clear justification for your work.

5. Conclusion

In the conclusion, please show how the work advances the field from the present state of knowledge. Please provide a clear justification for your work in this section, and indicate uses and extensions if appropriate. Moreover, you can suggest future experiments/simulations and point out those that are underway. The conclusion section has to be rewritten doing an effort to remark the main findings rather than summarizing the article content.

Author Response

(The authors gave the same response as above.)

Round 2

Reviewer 2 Report

The authors have improved the work according to the suggestions and recommendations made, answering point by point. It can be appreciated that the work can be published in an improved form.